# Understanding the implication of direct health facility financing on health commodities availability in Tanzania

George M. Ruhago[1]*, Michael B. John[2], Frida N. Ngalesoni[1], Daudi Msasi[3], Ntuli Kapologwe[4], James T. Kengia[4], Elias Bukundi[5], Regina Ndakidemi[6], Mavere A. Tukai[2]

1 Department of Development Studies, School of Public Health and Social Sciences, Muhimbili University of Health and Allied Sciences, Dar es Salaam, Tanzania, 2 USAID Global Health Supply Chain Program Technical Assistance, Dar es Salaam, Tanzania, 3 Ministry of Health, Community Development, Gender, Elderly and Children, Dodoma, Tanzania, 4 President's Office Regional Administration and Local Government (PORALG), Dodoma, Tanzania, 5 Department of Epidemiology and Biostatics, School of Public Health and Social Sciences, Muhimbili University of Health and Allied Sciences, Dar es Salaam, Tanzania, 6 Tanzania Public Sector Systems Strengthening Plus (PS3+) Project, Dar Es Salaam, Tanzania

* ruhagogm@gmail.com

**Data Availability Statement:** All data will be made available within the manuscript

## Abstract

The Government of Tanzania (GoT) has in the last decade made progress in strengthening the health system financing with progress towards Universal Health Coverage (UHC). The major reforms includes development of the health financing strategy, reforming the Community Health Fund (CHF) and introduction of the Direct Health Facility Financing (DHFF). DHFF was introduced in all district councils in the 2017/18 financial year. One of the anticipated goals of DHFF is to improve availability of health commodities. The objective of this study is to assess the effect of DHFF in improving the availability of health commodities in primary health care facilities. This study employed cross sectional study design, using quantitative techniques to analyze data related to expenditures and availability of health commodities at the primary health care facilities in Tanzania mainland. Secondary data was extracted from Electronic Logistics Management Information System (eLMIS) and Facility Financial Accounting and Reporting System (FFARS). Descriptive analysis was used to summarize the data using Microsoft Excel (2021) and inferential analysis was done using Stata SE 16.1. There has been an increase in allocation of funds for health commodities over the past three years. The Health Basket Funds (HBFs) accounted for an average of 50% of all health commodities expenditures. The complimentary funds (user fees and insurance) contributed about 20%, which is less than the 50% required by the cost sharing guideline. There is potentiality in DHFF improving visibility and tracking of health commodities funding. Implementation of DHFF has increased the amount of funding for health commodities at health facilities. The visibility and tracking of health commodity funding has improved. There is a scope of increasing health commodity funds at health facilities since the expenditures on health commodities is lower than what is indicated in the cost sharing collection and use guideline.

**Funding:** USAID- Global Health Supply Chain – Technical Assistance Tanzania for financial support of this work. But The funders had no role in study design, data collection and analysis, decision to publish, or preparation of the manuscript.

**Competing interests:** The authors have declared that no competing interests exist.

**Abbreviations:** CCHP, Comprehensive Council Health Plans; CHF, Community Health Fund; DHFF, Direct health Facility Financing; eLIMIS, Electronic logistic management information system; MSD, Medical Stores Department; FFRAS, Facility Financial Accounting and Reporting System; GFS, Government Finance Statistics; GOT, Government of Tanzania; HSBF, Health Sector Basket Fund; NHIF, National Health Insurance Fund; RBF, Result Based Financing.

# Introduction

To accelerate the progress towards achieving Universal Health Coverage (UHC), The Government of Tanzania (GoT) is committed to improve delivery of quality and equitable health services across all levels of service delivery [1]. Health services delivery in Tanzania is overseen by two key ministries. The Ministry of Health (MoH) [2] is involved with the formulation of health policy, guidelines, strategies, and resource mobilization. The President's Office of Regional Authority and Local Government (PORALG) oversees the delivery of Primary Health Care (PHC) i.e. dispensaries, health centers and district hospitals. Prior to 2017/18, funds for health services delivery at primary health care were channeled through the districts headquarters who would allocate and spend the funds on behalf of health facilities. In 2017/18, a new policy of Direct Health Financing was instituted. All funds for service delivery for PHC are allocated directly to health facilities from the Ministry of Finance and Planning (MoFP) [3].

The GOT has in the last decade made progress in strengthening the health financing towards UHC. The major efforts in health financing includes the development of the health financing strategy, reforming the community health fund (CHF) and introduction of the direct health facility financing (DHFF), in all district councils in 2017/18 [3–5]. The DHFF is a fiscal decentralization of financial resources directly to health facilities to improve health system performance by linking payment to priority service, enhancement of autonomy, transparency, and accountability at the facility level. With the introduction of DHFF, Health Basket Funds (HBFs) complimentary funds that includes user fees, Community Health Funds (CHF), National Health Insurance Fund (NHIF) and Results-Based Financing (RBF) funds, are now channeled directly to individual health facilities' bank accounts and utilized as planned.

To enable the operationalization of the DHFF the GoT introduced Facility Financial and Accounting System (FFARS) an electronic solution for improving financial management at facilities level [4]. FFARS is an information financial management system used for accounting and reporting of funds disbursed directly to health facility bank accounts [5]. The introduction of DHFF and the facilitating systems such as FFARS intends to improve efficiency in planning, budgeting, monitoring and evaluation of resource utilization, including procurement and distribution of health commodities. Health commodities are items required for the delivery of quality health services, including medicines, vaccines, laboratory/diagnostic consumables and medical supplies such as contraceptives, dressings, needles and syringes [6].

The DHFF policy introduction has enabled health facilities to plan and budget according to their demands. The health facilities have autonomy in procurement of health commodities [3]. In addition to implementing DHFF and FFARS, the GoT has also developed web-based planning and budgeting tool known as PlanRep to assist in strategic planning and resource allocation. Further the Comprehensive Council Health Planning (CCHP) and Health Facilities Planning guidelines has been developed to facilitate resource allocation including health commodities [7, 8]. It is envisaged that the reforms will increase provider autonomy over access to and use of resources, increase engagement of health facility governing committees and health care workers in the planning and financing of necessary inputs such as health commodities essential for the delivery of quality health services [3, 8].

Health financing arrangements geared towards ensuring availability of adequate financial resources and efficiency in resource utilization is crucial in achieving intended goals of UHC [9, 10]. According to the World Health Organization [11] the inefficient use of resources hinders rapid movement towards UHC [12, 13]. Efficient resource use reduces waste, enhances the ability of health systems to provide quality services and improve population health. The availability of essential health commodities is core for any health system working to achieve

UHC [14]. Strong health systems are advocated as a tool for accelerating UHC [15]. The current improvement in the financial management information systems in Tanzania is key in the allocation and tracking of financial resources for essential health commodities towards UHC. Equitable access to essential health commodities is an important component for a well-functioning health system [14, 16]. Monitoring and evaluation of reforms that are pitched on improving availability of health commodities is key towards realization of the UHC and safeguarding equity in access and use of health services. However, to date, there is paucity of documented literature in Tanzania with regards to the effectiveness of DHFF reforms in improving the availability of health commodities in primary health care facilities.

The objective of this study is therefore to assess the effect of health system financing reforms such as DHFF in improving the availability of health commodities in primary health care facilities. Specifically, the study aims at assessing the effect of DHFF on health commodities availability and expenditure trends related to health commodities in the context of DHFF. The findings of this study will provide recommendations for further improvements of the health commodities supply chain in Tanzania.

## Methods

### Study area and design

Tanzania is a lower middle-income country located in East Africa, with a population of 61,627,284 and an estimated area of 945,087 km$^2$. Administratively, Tanzania mainland has 26 regions, 139 Districts, 185 Councils, 570 Divisions, 3956 Wards and 12319 villages [17]. Tanzania has had a steady economic growth ranging from 5% to 7% prior to covid-19, but fell to 2.1% in 2020 from 6.8% in 2019, currently estimated at 4.5% to 5% [18]. This study was conducted in the context of understanding the effect of DHFF in improving the availability of health commodities in the primary health care settings in Tanzania. The health system in Tanzania operates in a decentralized system. Under the decentralized systems, three functional levels exist namely: district (primary level), regional (secondary level), and referral hospital (tertiary level). By January 2021, there were a total of 5,865 registered public primary health care facilities out of which 5084 were Dispensaries, 639 Health Centers and 142 District Hospitals [19].

This study employed cross sectional study design, using quantitative techniques to analyzed data on health commodity expenditure and health commodities availability at the primary health care facilities in Tanzania mainland.

### Sampling procedures and sample size

A total of 5,143 health facilities across 26 regions in Tanzania mainland whose expenditure data were available in FFARS were included in the analysis, among which 83 were district hospitals, 530 were health centers, and 4,530 were dispensaries. Whilst the data for the health commodities list was pulled from eLIMIS, the system provides reports such as stock on hand, stock out days consumption, losses, and adjustments. The platform is also used for ordering health commodities [20].

### Data collection tools and procedures

The data was extracted from two different management information systems namely eLMIS for health commodity data and FFARS for financial resources utilized to procure health commodities at health facilities.

## Data management and analysis

The key variables of interest for health commodity financing included sources of funds and expenditures by categories based on charts of accounts. The expenditures on procurement of health commodities were analyzed using data for financial years 2017/18, 2018/19. 2019/20. 2020/21 extracted from FFARS. The eLMIS database, provided data on the supply levels of 312 health commodities from MoH essential commodities tracer list. The analysis for health commodities included 2016/17, 2017/18, 2018/19, 2019/20 and 2020/21. The year 2016/17 was used as baseline before the launch of DHFF. However, baseline for expenditures was set at 2017/18 when DHFF commenced. MoH essential commodities tracer list was used to measure annual availability of health commodities at health facilities. Descriptive statistics was used to summarize the utilization of health commodity expenditures and health commodities availability at the primary health care. We plotted graphs to show aggregate trend of health commodities expenditure and availability of by facility type. Further two sample test of proportion was used to identify if there was any significant difference in the health commodity availability between rural and urban area.

## Results

### Health commodity expenditure

Comparing the health facility expenditure as a percentage of the total expenditures, it was revealed that for the last four financial years from 2017/18 to 2020/21on average health facilities allocated about 23 percent of their budget to procure health commodities. The health commodities expenditure increased steadily from 22.5 percent of the total health expenditure in 2017/18 to 24.7 percent in 2019/20. However, in 2020/21 declined to 22.5 percent of the total health expenditures. (Table 1).

### Expenditure on health commodities by facility type

Expenditures on health commodities at the facility level revealed that, council hospital over the four years 2017/18 to 2020/21, on average spent about 30 percent of their health expenditure on health commodities compared to 22.5 and 23.1 percent for health center and dispensaries respectively (Table 2).

### Health commodity expenditure by geographical location

Examining health commodities expenditure by health facility location revealed that, on average urban health facilities were likely to spend more of their budget on health commodities 25.6 percent compared to 22.6 percent in rural area facilities. However, looking on the trend overtime it can be observed that, expenditure on health commodities have been increasing in rural area facilities from USD$ 12.1 Mil (20.2% of the total health expenditure) in 2017/18 to USD$ 13.5Mil (23.3% of the total health expenditure) in 2020/21 (Table 3).

**Table 1. Health commodities expenditure as a percentage of total health facilities expenditures (USD$).**

| Category | 2017/2018 | 2018/19 | 2019/20 | 2020/21 |
|---|---|---|---|---|
| Health commodities expenditure | 17,600,000 | 21,400,000 | 24,600,000 | 23,200,000 |
| Other health services expenditure | 60,800,000 | 70,300,000 | 74,900,000 | 79,100,000 |
| **Total expenditure** | **78,500,000** | **91,700,000** | **99,500,000** | **102,000,000** |
| **Other health expenditure % of total expenditure** | **77.5%** | **76.7%** | **75.3%** | **77.5%** |
| **Health commodities expenditure % of total expenditure** | 22.5% | 23.3% | 24.7% | 22.5% |

**Table 2. Expenditure on health commodities by facility type (USD$).**

| Facility Type | 2017/18 | 2018/19 | 2019/20 | 2020/21 |
|---|---|---|---|---|
| **Council Hospital** | | | | |
| Health commodities expenditure | 4,777,038 | 6,456,518 | 7,227,727 | 9,017,188 |
| Other health services expenditure | 7,965,797 | 13,100,000 | 19,800,000 | 31,300,000 |
| Total expenditure | 12,742,835 | 19,556,518 | 27,027,727 | 40,317,188 |
| *% health commodities expenditure over total expenditure* | **37.5%** | **33.0%** | **26.7%** | **22.4%** |
| **Health Center** | | | | |
| Health commodities expenditure | 5,858,808 | 7,091,521 | 7,971,271 | 6,123,707 |
| Other health services expenditure | 32,400,000 | 30,700,000 | 21,900,000 | 19,200,000 |
| **Total expenditure** | 38,258,808 | 37,791,521 | 29,871,271 | 27,171,271 |
| *% Health commodities expenditure over total expenditure* | **15.3%** | **18.8%** | **26.7%** | **29.3%** |
| **Dispensary** | | | | |
| Health commodities expenditure | 7,007,767 | 7,860,217 | 9,413,928 | 8,079,316 |
| Other health services expenditure | 20,400,000 | 26,600,000 | 33,300,000 | 28,600,000 |
| **Total expenditure** | 27,407,767 | 34,460,217 | 42,713,928 | 36,679,316 |
| *% Health commodities expenditure over total expenditure* | **25.6** | **22.8** | **22.0** | **22.0** |

## Expenditure on health commodities by source of fund facility type

Disaggregating the health commodity funds by the source of fund indicate that HBSF is the largest source of health commodity funding contributing about 63% of all the expenditures in 2017/18, however that declined to 45.7% in 2020/21. User fees is increasingly becoming the important source of funding for health commodities contributing about 31.7% of the source of funding for 2020/2021 from 10.5% in 2017/18 (Table 4).

## Health commodity availability of the 312 tracer medicine by facility type

The MoH essential commodities tracer list of 312 health commodities was analyzed from eLIMIS. At the start of DHFF implementation, the average health commodity availability was 69%. After DHFF implementation, this increased to a peak of 78% in 2018/19. (Fig 1).

## Health commodity availability by facility level

Based on the available data we examined health commodities availability by facility level between 2018/19 to 2020/2021. The availability of health commodities measured by the 312

**Table 3. Health commodity expenditure by geographical Location (USD$).**

| Geographical location | 2017/2018 | 2018/2019 | 2019/2020 | 2020/2021 |
|---|---|---|---|---|
| **Rural** | | | | |
| Health commodities expenditure | 12,100,000 | 14,500,000 | 16,600,000 | 13,500,000 |
| Other health services expenditure | 47,800,000 | 50,400,000 | 50,800,000 | 44,400,000 |
| Total expenditure | 59,900,000 | 64,900,000 | 67,400,000 | 57,900,000 |
| *% Health commodities expenditure over total expenditure* | **20.2%** | **22.3%** | **24.6%** | **23.3%** |
| **Urban** | | | | |
| Health commodities expenditure | 5,518,566 | 6,956,902 | 8,008,164 | 9,757,580 |
| Other health services expenditure | 13,000,000 | 19,900,000 | 24,100,000 | 34,700,000 |
| **Total expenditure** | 18,518,566 | 26,856,902 | 32,108,164 | 44,457,580 |
| *% Health commodities expenditure over total expenditure* | **29.8%** | **25.9%** | **24.9%** | **21.9%** |

**Table 4. Health commodities expenditure by source of fund (USD$).**

| Category | HSB | Insurance | DRF | User Fees | Own Sources | NGO | Central government | Donors | Others | Total |
|---|---|---|---|---|---|---|---|---|---|---|
| **2017/2018** | | | | | | | | | | |
| Health commodities expenditure | 11,000,000.0 | 3,374,019.0 | 240,151.9 | 1,854,850.0 | 333,742.3 | 130.4 | 150,609.4 | 646,065.6 | 217.4 | 17,599,786.0 |
| % | **62.5%** | **19.2%** | **1.4%** | **10.5%** | **1.9%** | **0.0%** | **0.9%** | **3.7%** | **0.0%** | **100%** |
| **2018/2019** | | | | | | | | | | |
| Health commodities expenditure | 10,900,000.0 | 6,033,663.0 | 714,680.0 | 2,781,708.0 | 47,579.0 | 50,020.0 | 126,917.3 | 787,380.4 | 645.0 | 21,442,592.7 |
| % | **50.8%** | **28.1%** | **3.3%** | **13.0%** | **0.2%** | **0.2%** | **0.6%** | **3.7%** | **0.0%** | **100.0%** |
| **2019/2020** | | | | | | | | | | |
| Health commodities expenditure | 12,200,000.0 | 4,953,527.0 | 754,935.1 | 4,734,692.0 | 92,401.7 | 44,633.8 | 27,165.9 | 1,744,470.0 | 33,052.2 | 24,584,877.7 |
| % | **49.6%** | **20.1%** | **3.1%** | **19.3%** | **0.4%** | **0.2%** | **0.1%** | **7.1%** | **0.1%** | **100.0%** |
| **2020/2021** | | | | | | | | | | |
| Health commodities expenditure | 10,600,000.0 | 3,799,306.0 | 1,126,418.0 | 7,341,038.0 | 37,700.0 | 11.8 | 18,038.7 | 228,985.1 | 29,558.4 | 23,181,056.0 |
| % | **45.7%** | **16.4%** | **4.9%** | **31.7%** | **0.2%** | **0.0%** | **0.1%** | **1.0%** | **0.1%** | **100.0%** |

tracer medicine in 2018/19 was above the national target of 80% at hospital level, at health center and dispensary the availability was closer to the target. However, at all levels, the availability decreased in the second and third year of analysis. The decline was at different levels, this suggests that DHFF may have affected commodity availability differently at different types of facilities. (Table 5 and Fig 2).

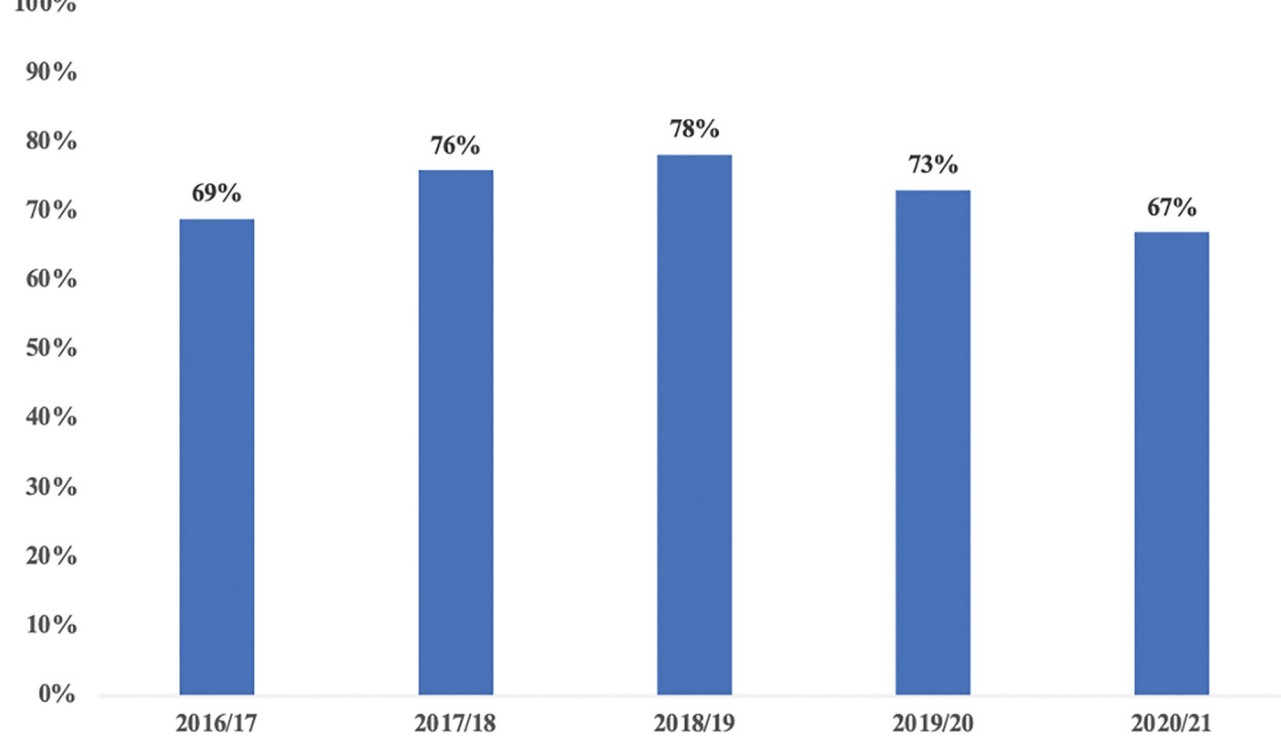

**Fig 1. Average health commodity availability of the 312 tracer medicine list.**

**Table 5. Health commodity availability by facility type (USD$).**

| Facility Type | 2018/19 | 2019/20 | 2020/21 |
|---|---|---|---|
| **Council Hospital** | | | |
| Health commodities expenditure | 6,456,518 | 7,227,727 | 9,017,188 |
| Total expenditure | 19,556,518 | 27,027,727 | 40,317,188 |
| *% health commodities expenditure over total expenditure* | **33.0%** | **26.7%** | **22.4%** |
| *Average availability of medicines* | **81.7%** | **71.4%** | **64.3%** |
| **Health Center** | | | |
| Health commodities expenditure | 7,091,521 | 7,971,271 | 6,123,707 |
| **Total expenditure** | 37,791,521 | 29,871,271 | 27,171,271 |
| *% Health commodities expenditure over total expenditure* | **18.8%** | **26.7%** | **29.3%** |
| *Average availability of medicines* | **79.6%** | **74.2%** | **68%** |
| **Dispensary** | | | |
| Health commodities expenditure | 7,860,217 | 9,413,928 | 8,079,316 |
| **Total expenditure** | 34,460,217 | 42,713,928 | 36,679,316 |
| *% Health commodities expenditure over total expenditure* | **22.8** | **22.0** | **22.0** |
| *Average availability of medicines* | **77.9%** | **72.5%** | **54.8%** |

## Health commodity availability urban vs rural health facilities

A comparison of health commodities availability by the geographical location of the health facility revealed that urban facilities were more likely to have health commodities in financial years 2019/20 and 2020/21, the difference was statistically significant in 2020/2021, P value = 0.044 (Table 6).

## Discussion

This study findings indicate that there is potentiality of DHFF improving availability of health commodities funds at the health facilities. In this study it has been revealed that, the largest funding source is Health Basket Funds (HBFs), which account for an average of 52% of all health facility funds. HSBF, insurance funds i.e., (the National Health Insurance Funds (NHIF) reimbursements, Improved Community Health Funds (iCHF)), user fees, and Drug Revolving Fund (a central government funding for health commodities), when added together,

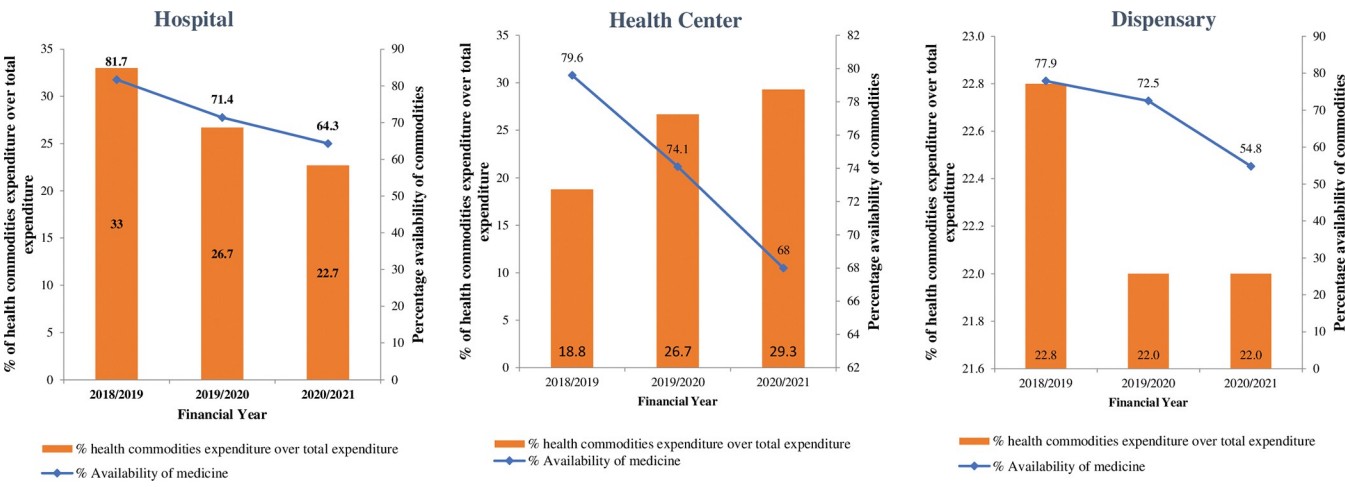

**Fig 2. Aggregate trend of health commodities expenditure and availability by health facility levels.**

**Table 6. Health commodity availability urban vs rural health facilities.**

| Year | Rural | | Urban | | P-value |
|---|---|---|---|---|---|
| | Number of health facilities | Percentage availability of health commodities | Number of health facilities | Percentage availability of health commodities | |
| 2018/2019 | 3,007 | 78.1 | 443 | 80.2 | 0.316 |
| 2019/2020 | 3007 | 71.3 | 443 | 75.7 | 0.054 |
| 2020/2021 | **3007** | **63.9** | **443** | **68.8** | **0.044** |

account for over 90% of total health commodities expenditures at health facilities. However, over the past two financial years user fees has become a significant source of funding increasing by about three folds. Towards universal health coverage includes reduction on over reliance to out-of-pocket financing [21]. The observed increase in out of pocket expenditure could be attributed to declining development assistance for health in Tanzania and other developing countries [22]. This calls for concerted efforts to improve financial protection by reducing household out-of- pocket spending on health towards universal health coverage.

To improve the efficiency in spending, increase accountability, and monitor the effect of expenditures on health requires a close management and supervision [23] The Government of Tanzania has developed guideline on the utilization of complimentary funds i.e. funds collected at the facility level such as user fees and insurance funds (NHIF, iCHF), recommends that at least 50% of the fund should be utilized to procure health commodities [2]. This study reported that health facilities allocated, on average, about 20% of their available health budget on health commodities, compared to the guideline recommendation of spending a minimum of 50% to procure health commodities. The low adherence to cost sharing guidelines calls for further studies to better understand whether provisions of the guideline is sufficient, or if there are any other health facility priorities that are not reflected in the guidelines. A recent study in Tanzania has indicated that lack of well-tailored governance mechanism hampers the implementation of health financing reform mechanism. The identified barriers included; a lack of transparency, limited and weak accountability for revenues generated from health care facilities [24].

Looking at the absolute term, health commodity expenditure increased more for urban area compared to rural area. Though not analyzed by this study, this could partly be explained by the covid 19 pandemic. The pandemic might have forced the health system to spend funding elsewhere, such as health education and preventive services e.g., WASH interventions. This intervention targeted urban areas due the high prevalence rate of covid19 in urban areas compared to rural areas. Future studies could focus on this area to explore the equity in financing between rural and urban areas.

Similar trends are observed, when examined by level of health services. The funding is steadily increasing at hospital levels but fluctuating at health centers and dispensaries. It remains an empirical question that needs further exploratory studies to identify the underlying causes. One underlying cause could be inadequate or low skilled accountants in lower-level facilities leading to poor data quality. A previous study on the review of the implementation of FFARS in Tanzania [5] indicated existence of inadequate accountants, one district had only four accountants, that oversees about 100 facilities. Another study in Tanzania on the use Plan-Rep a web based planning and budgeting tools, indicated that shortage of ICT equipment and access to reliable internet complicates the utilization of such innovative systems, more so in rural areas [4].

The findings of this study have indicated that during the same period of implementing DHFF, there has been increased availability of health commodities measured by tracer medicine. Although there are other factors which might have contributed to this change. A study in Cameroon corroborated, whereby increased financing coupled with facility autonomy was related to perceived increase in health commodity availability [25]. However, in 2019/20 and 2021/22 there was a slight decline in the availability of health commodities. The observed trend, might be associated with covid19 pandemic in which resources where diverted to covid19 related interventions [26, 27].

The interpretation of the results of this study should consider some of the limitations. This paper considered spending of funds received directly into the bank accounts through DHFF. However, there are some of the health commodities funds, such as Receipt In-Kind (RIK) or Drug Revolving Fund (DRF) that are disbursed by the central government into MSD accounts and the facilities are supplied with health commodities [28]. This important funding might not have been captured accurately in this study. A study on health facility financial needs in Tanzania, indicated that RIK or DRF was an important source of financing for primary health facilities. Therefore, while interpreting the findings of this study it is important that such factors are taken into consideration that some of key source of health commodities funding might not have been captured in its entirety in this study. There was no financial data by facility levels prior to DHFF in 2016/17, this limits the discussion of the effect of the health financing prior to DHFF policy. This paper has only analyzed expenditure data on health commodities, the inconsistencies in transferring budget data from PlanRep to FFARS due to the ongoing interoperability bottlenecks, did not allow incorporating budget and disbursement data in the analysis [4].

## Conclusions

The implementation of DHFF has increased availability of health commodities financing at health facilities. Improved visibility and tracking expenditures on health commodity. However, there is a notable urban rural difference in health commodities financing. Availability of health commodities were stable in urban areas compared to rural areas. There is a scope of increasing health commodity funds at health facilities since the amount spend on health commodities was lower than what is indicated in the cost sharing guideline.

## Acknowledgments

The authors would like to thank PORALGA and MoH for allowing us to use FFARS and eLIMIS data in this study.

## Author Contributions

**Conceptualization:** George M. Ruhago, Michael B. John.

**Data curation:** George M. Ruhago.

**Formal analysis:** George M. Ruhago, Elias Bukundi.

**Methodology:** George M. Ruhago, Michael B. John, Frida N. Ngalesoni, Ntuli Kapologwe, James T. Kengia, Regina Ndakidemi, Mavere A. Tukai.

**Supervision:** Mavere A. Tukai.

**Validation:** Daudi Msasi, Ntuli Kapologwe, James T. Kengia, Regina Ndakidemi, Mavere A. Tukai.

**Writing – original draft:** George M. Ruhago, Michael B. John.

**Writing – review & editing:** George M. Ruhago, Michael B. John, Frida N. Ngalesoni, Daudi Msasi, Ntuli Kapologwe, James T. Kengia, Elias Bukundi, Regina Ndakidemi, Mavere A. Tukai.

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
