## [Decision Letter · Decision Letter 0]

10 Jun 2022

PGPH-D-22-00644

Understanding the implication of Direct Health Facility Financing on Health Commodities availability in Tanzania

Dear Dr. Ruhago,

Thank you for submitting your manuscript to PLOS Global Public Health. After careful consideration, we feel that it has merit but does not fully meet PLOS Global Public Health’s publication criteria as it currently stands. Therefore, we invite you to submit a revised version of the manuscript that addresses the points raised during the review process.

Please submit your revised manuscript by . If you will need more time than this to complete your revisions, please reply to this message or contact the journal office at globalpubhealth@plos.org. Please include the following items when submitting your revised manuscript:

We look forward to receiving your revised manuscript.

Kind regards,

Cynthia Chen

Academic Editor

Journal Requirements:

1. Please send a completed 'Competing Interests' statement, including any COIs declared by your co-authors. If you have no competing interests to declare, please state "The authors have declared that no competing interests exist". 

a. Please clarify all sources of funding (financial or material support) for your study. List the grants (with grant number) or organizations (with url) that supported your study, including funding received from your institution. 

b. State the initials, alongside each funding source, of each author to receive each grant.

c. State what role the funders took in the study. If the funders had no role in your study, please state: “The funders had no role in study design, data collection and analysis, decision to publish, or preparation of the manuscript.”

3. In the online submission form, you indicated that "Data will be available from authors upon request". All PLOS journals now require all data underlying the findings described in their manuscript to be freely available to other researchers, either 1. In a public repository, 2. Within the manuscript itself, or 3. Uploaded as supplementary information.

4. Your manuscript is missing the following sections: Introduction section. Please ensure these are present, and in the correct order, and that any references to subheadings in your main text are correct. An outline of the required sections can be consulted in our submission guidelines here: 

https://journals.plos.org/globalpublichealth/s/submission-guidelines#loc-parts-of-a-submission

Additional Editor Comments (if provided):

Reviewers' comments:

Reviewer's Responses to Questions

**Comments to the Author**

1. Does this manuscript meet PLOS Global Public Health’s publication criteria? Is the manuscript technically sound, and do the data support the conclusions? The manuscript must describe methodologically and ethically rigorous research with conclusions that are appropriately drawn based on the data presented.

Reviewer #1: Partly

Reviewer #2: No

2. Has the statistical analysis been performed appropriately and rigorously?

Reviewer #1: Yes

Reviewer #2: No

3. Have the authors made all data underlying the findings in their manuscript fully available (please refer to the Data Availability Statement at the start of the manuscript PDF file)?

Reviewer #1: No

Reviewer #2: No

4. Is the manuscript presented in an intelligible fashion and written in standard English?

Reviewer #1: Yes

Reviewer #2: Yes

5. Review Comments to the Author

Reviewer #1: This paper evaluates the impact of Direct Health Facility Financing on health commodities availability in Tanzania since the implementation of DHFF in 2017/18. The results show that average health commodity availability increased from 64% in 2015/16 to 78% in 2018/19 and declined in later years in response to decline in total health expenditure. This results are novel and has policy implications in improving health care financing in developing countries.

A few comments:

1. It would be helpful to explain how finance resources were allocated prior to DHFF in the introduction of the paper, so the policy implications of DHFF will be better understood for readers without background knowledge.

2.How was the key variable in Figure 1, Average Health Commodity Availability defined, is this index an average of the availability of all 312 health commodities?

3. In Figure 1, there was an increase in Average Health Commodity Availability from 2015/16 to 2016/17. A discussion on the pre-DHFF trend of health commodity availability would be helpful for the reader to decompose the policy effect of DHFF from the baseline increasing tread.

4. I recommend changing the type of the Figure 2 chart to a clustered column chart. The current format of the chart is misleading, the numbers does not sum up to 100% in each column.

Several minor comments:

1. The abbreviation "MSD" in the last paragraph on Page 5 is not included in the List of Abbreviations, or explained in the article.

2. What is the currency unit in Table 1?

In addition, I would recommend checking the grammar. For example, on Page 11, the second paragraph, a period is needed after the first sentence. In the same paragraph, the phrase "The low adherence to health commodity cost sharing guidelines" is not a complete sentence.

Reviewer #2: This study explored an important topic on strengthening health system financing and universal health coverage. It aimed to examine the effect of DHFF in improving the availability of health commodities in primary health care facilities in Tanzania.

However, there are some major issues in this work including weak study design and unreliable statistical method. The main results of the study was derived using Excel to summarize the health commodity budgeting, expenditure and availability on yearly base. No statistical inference can be derived. In another way, readers cannot draw convincing conclusion from the results. There were also not enough background information, theoretical setup and discussion for the readers to understand how the DHFF solve the existing problem and how it can facilitate the road towards UHC in Tanzania.

My detailed comments and suggestions are below.

Background:

The authors should provide more information on the current healthcare system in Tanzania and the existing challenges/issues in the healthcare system. In particular, the issues related to health care financing should be explained. This can help readers better understand the rationale of introducing DHFF.

Method:

The method of the study is my major concern. The study aimed to examine the effect of DHFF. However, the method only involves summarizing data on yearly base. The readers cannot draw robust conclusion from the results.

What are the confounding factors and how do these factors affect the results? For example, FFARS was also introduced in parallel to DHFF. How does this affect the interpretation of the results?

Are the effects statistically significant?

The statement in Data management and analysis section is inconsistent. I pasted the sentence below:

Data from four years was analyzed: one year before the implementation of DHFF (2015/16) and three years after (2016/17, 2017/18, 2018/19, and 2019/20).

One suggestion is to consider regression analysis by considering the data at sub-region level: e.g.: 26 regions, 139 Districts, 185 Councils, 570 Divisions, 3956 Wards or 12319 villages.

Further, the authors should consider collecting and showing data before 2015/16 to examine the trend in the outcome variables.

The authors should explain more about health commodities. This is not intuitive to me. What do health commodities include?

Measure for "health commodities availability" was not well defined.

Results:

The main results are summary statistics at yearly base. I cannot draw any statistical inference.

Table 1: there were huge reductions in health commodities budget from 2018/2019 to 2019/2020. In my opinion, the results “With the overall spending for 2017/18 being 20% of the total budgets while 27%, 80% and 96% for 2020/21” was mainly driven by a reduction in budget. I am not sure how to interpret the results. Does this mean inefficient planning at the beginning?

Similar comments apply to Table 2.

Furthermore, were all the budget used efficiently? The authors mentioned about adherence to expenditure guidelines. But, more information and discussion should be provided.

Table 3: this table is difficult to read. Please consider to change the format or use a figure to show the trend.

Figure 1: "health commodity availability" was not well defined. The authors claimed “After DHFF implementation, this increased each year to 69% availability in 2016/17 and 76% availability in 2017/18.” It seems to me that the authors wanted to claim a casual impact here. But in my opinion, this statement is not meaningful. There was already an increase before the introduction of DHFF, from 64% to 69%. And furthermore, there could be other things happening at 2016/2017. This goes back to the issue of the statistical method.

Discussion

Because the statistical method is not robust, the results cannot support many of the claims in the Discussion.

Overall, the language is fine. But, additional language editing and proof reading are required as there are some typos and grammatical errors.

6. PLOS authors have the option to publish the peer review history of their article (what does this mean?). If published, this will include your full peer review and any attached files.

**Do you want your identity to be public for this peer review?** For information about this choice, including consent withdrawal, please see our Privacy Policy.

Reviewer #1: No

Reviewer #2: No

---

## [Decision Letter · Decision Letter 1]

28 Dec 2022

PGPH-D-22-00644R1

Understanding the implication of Direct Health Facility Financing on Health Commodities availability in Tanzania

Dear Dr. Ruhago,

Thank you for submitting your manuscript to PLOS Global Public Health. After careful consideration, we feel that it has merit but does not fully meet PLOS Global Public Health’s publication criteria as it currently stands. Therefore, we invite you to submit a revised version of the manuscript that addresses the points raised during the review process.

We look forward to receiving your revised manuscript.

Kind regards,

Cynthia Chen

Academic Editor

Journal Requirements:

2. Please send a completed 'Competing Interests' statement, including any COIs declared by your co-authors. If you have no competing interests to declare, please state "The authors have declared that no competing interests exist". Otherwise please declare all competing interests beginning with the statement "I have read the journal's policy and the authors of this manuscript have the following competing interests:"

3. Your manuscript is missing the following sections: Introduction. Please ensure these are present, and in the correct order, and that any references to subheadings in your main text are correct. An outline of the required sections can be consulted in our submission guidelines here:

https://journals.plos.org/globalpublichealth/s/submission-guidelines#loc-parts-of-a-submission

Additional Editor Comments (if provided):

Dear author,

There are suggestions and inconsistencies highlighted by the reviewers that will be helpful to address. If data is available, it will be helpful to analyze data from health facility instead of aggregated data.

Reviewers' comments:

Reviewer's Responses to Questions

**Comments to the Author**

1. If the authors have adequately addressed your comments raised in a previous round of review and you feel that this manuscript is now acceptable for publication, you may indicate that here to bypass the “Comments to the Author” section, enter your conflict of interest statement in the “Confidential to Editor” section, and submit your "Accept" recommendation.

Reviewer #1: All comments have been addressed

Reviewer #2: All comments have been addressed

2. Does this manuscript meet PLOS Global Public Health’s publication criteria? Is the manuscript technically sound, and do the data support the conclusions? The manuscript must describe methodologically and ethically rigorous research with conclusions that are appropriately drawn based on the data presented.

Reviewer #1: Yes

Reviewer #2: Yes

3. Has the statistical analysis been performed appropriately and rigorously?

Reviewer #1: No

Reviewer #2: No

4. Have the authors made all data underlying the findings in their manuscript fully available (please refer to the Data Availability Statement at the start of the manuscript PDF file)?

Reviewer #1: No

Reviewer #2: Yes

5. Is the manuscript presented in an intelligible fashion and written in standard English?

Reviewer #1: Yes

Reviewer #2: No

6. Review Comments to the Author

Reviewer #1: Thank you for addressing my comments in the revised version. I have minor concerns on the statistical analysis in Figure 2 and Table 6.

Figure 2: Using regression analysis in the revised is not necessary here. There are only 3 observations in each graph so there is very limited statistical power, the R-squared and coefficients does not mean a lot. In addition, these observations are time-series data, so the independence assumption of linear regression is not satisfied: health care facilities can budget their expenditure and medicine purchase so the data points are not independent observations. Further more, only the correlation was discussed in the manuscript, and the regression is not explained in the text.

My recommendation is to choose from one of the two following options:

1. report the aggregate trend by facility type, do not use regression analysis

2. If you prefer regression analysis, the observations should be individual health facilities. Then you have 5,143 facilities times 3 years = 15,429 observations. The independent variables can be year dummies, facility type, urban/rural, etc.

Table 6: Please report number of Rural and Urban health facilities in the table.

Reviewer #2: Appreciate the efforts and the additional works from the authors.

My detailed comments are below:

Minor comments:

Please do another round of proofreading. I can still identify many typos and grammar errors, which affect the readability of the manuscript. I give some examples below.

“The complimentary funds (user fees and insurance) contributed about 20%) less than the 50% required by the cost sharing guideline.”

Extra “)”.

“In 2017/18 a new policy of Direct Health Financing was instituted,”

When a time phrase adds information to an independent clause or sentence that follows it, it should be followed by a comma.

The next sentence is an independent sentence. Full stop should be used after this sentence.

“In addition to implementing DHFF and FFARS, the Government, has also developed web based planning and budgeting tools PlanRep to assist in strategic planning and resource allocation.”

Extra comma after “the Government”.

“Strong health systems are advocated as a tool for accelerating UHC.(13)”

Citation should be before the full stop.

“The availability of health commodities in 2018/19 was above the national target of 80% availability of tracer medicine at hospital level, at health center and dispensary the availability was closer to the target However, at all levels, the availability decreased in the second and third year of analysis.”

Misuse of and missing full stop.

Stata, not stata. Capital “S”.

“The President’s Office, Regional Administration and Local Government (PORALG) is responsible for service delivery in primary health care facilities through…”

PORALG has been explained in the Background. No need to repeat.

Explain “eLIMIS” at its first appearance in section "Sampling procedures and sample size"

Major comments:

Under "Data management and analysis", please provide more information about correlation analysis and regression analysis.

What tests were used for correlation analysis?

For regression analysis, what were the control variables/confounders? Were linear regression used or other type of regression used?

Can I consider introducing FFARS as part of the DHFF reform? Or FFARS is something independent, but happened to be implemented at the same time with DHFF?

Table 2: any explanation why “other health service expenditure” increased for Council Hospital but decreased for Health Center?

Table 3: looking at the absolute term, expenditure increased more for urban area compared to rural area. Why? Was rural area prioritized at the beginning?

“facilities from USD$ 12.1 Mil (22.2% of the total health expenditure) in…”

Typo? Should be 20.2%.

For the following results:

“Further examination of the correlation between health commodities availability and availability of funds, revealed that, there was a very strong positive correlation between percentage of health commodities expenditure and availabilities of medicines at the hospital level (R=1) and dispensary (R= 0.68). However, a negative correlation was observed between percentage of health expenditure on health commodities and availability of health commodities (R= - 0.94) (Figure 2).”

Is this part done by regression analysis? If this is from regression, please show the regression results in a table, including number of observations, coefficient, p-value, and 95% confidence interval. The authors can consider to put the Table in the Appendix/Supplementary Material.

Figure 2 is confusing. Based on Figure 2, it seems that the authors ran the regression just using 3 observations at the yearly level. Please clarify. The regression analysis should use the data at disaggregated level. It is uncommon that a regression analysis shows R-square = 1.

Are the results from Table 6 derived from t-test?

It seems counter intuitive that medicine availability decreased over time. I would expect DHFF increase the availability. Any reason?

Besides, how was health commodity availability calculated? For example, for medicine, is it that you take the ratio between the available medicine and the total number of medicine (312)? Same question applies to the calculation of health commodity availability.

Discussion:

The authors claimed that “This study findings indicate that there is potentiality in DHFF improving the availability of health commodities”. Which part of the results can support this? Is it because of the increased health expenditure over time? But health commodity availability decreased in the recent three years. Please clarify.

“The findings of this study have indicated that during the same period of implementing DHFF, there has been increased availability of health commodities measured by tracer medicine.”

This statement is inconsistent with results from Table 5.

7. PLOS authors have the option to publish the peer review history of their article (what does this mean?). If published, this will include your full peer review and any attached files.

**Do you want your identity to be public for this peer review?** For information about this choice, including consent withdrawal, please see our Privacy Policy.

Reviewer #1: No

Reviewer #2: No

---

## [Decision Letter · Decision Letter 2]

9 Mar 2023

PGPH-D-22-00644R2

Understanding the implication of Direct Health Facility Financing on Health Commodities availability in Tanzania

Dear Dr. Ruhago,

Thank you for submitting your manuscript to PLOS Global Public Health. After careful consideration, we feel that it has merit but does not fully meet PLOS Global Public Health’s publication criteria as it currently stands. Therefore, we invite you to submit a revised version of the manuscript that addresses the points raised during the review process.

We look forward to receiving your revised manuscript.

Kind regards,

Cynthia Chen

Academic Editor

Journal Requirements:

Additional Editor Comments (if provided):

Reviewers' comments:

Reviewer's Responses to Questions

**Comments to the Author**

1. If the authors have adequately addressed your comments raised in a previous round of review and you feel that this manuscript is now acceptable for publication, you may indicate that here to bypass the “Comments to the Author” section, enter your conflict of interest statement in the “Confidential to Editor” section, and submit your "Accept" recommendation.

Reviewer #1: All comments have been addressed

Reviewer #2: (No Response)

2. Does this manuscript meet PLOS Global Public Health’s publication criteria? Is the manuscript technically sound, and do the data support the conclusions? The manuscript must describe methodologically and ethically rigorous research with conclusions that are appropriately drawn based on the data presented.

Reviewer #1: Yes

Reviewer #2: Yes

3. Has the statistical analysis been performed appropriately and rigorously?

Reviewer #1: Yes

Reviewer #2: Yes

4. Have the authors made all data underlying the findings in their manuscript fully available (please refer to the Data Availability Statement at the start of the manuscript PDF file)?

Reviewer #1: No

Reviewer #2: Yes

5. Is the manuscript presented in an intelligible fashion and written in standard English?

Reviewer #1: Yes

Reviewer #2: Yes

6. Review Comments to the Author

Reviewer #1: (No Response)

Reviewer #2: Thanks for addressing my comments. I have some minor comments below.

In the Abstract – Results and the Main manuscript – Discussion, the authors mentioned “visibility” in the first sentence. This is confusing and not inline with the objective of the study. The objective of the paper is to understand the impact on health commodities availability. The author may mention “visibility” in the manuscript, but “visibility” should not be the first thing mentioned as this is not part of the study objective.

The authors are not obligated to say that the intervention increased commodity availability. The authors just need to reported the results.

Figure 1: the authors should remove data 2016/17. In the “Data management and analysis”, the authors mentioned data to be used are from 2017 to 2021. It is inconsistent to include data 2016/2017 in the figure. Alternatively, the authors should include a footnotes under Figure 1 to explain why there is data from 2016/2017 here.

The authors explained about the difference between rural areas and urban areas (Table 3 and Table 6) in their response letter but not in the manuscript. The authors should put this in the Discussion. This is important as potentially there is equity implication.

Table 5: the absolute Health commodities expenditure increased or increased first then decreased. However, the Average availability of medicine kept decreasing for all type of Facilities. The authors should discuss about this in the manuscript about the potential reasons and the implication on health.

There are still language and formatting issues. I will leave these to the journal editing team.

7. PLOS authors have the option to publish the peer review history of their article (what does this mean?). If published, this will include your full peer review and any attached files.

**Do you want your identity to be public for this peer review?** For information about this choice, including consent withdrawal, please see our Privacy Policy.

Reviewer #1: No

Reviewer #2: No

---

## [Editor Report · Decision Letter 3]

11 Apr 2023

Understanding the implication of Direct Health Facility Financing on Health Commodities availability in Tanzania

PGPH-D-22-00644R3

Dear Dr. Ruhago,

We are pleased to inform you that your manuscript 'Understanding the implication of Direct Health Facility Financing on Health Commodities availability in Tanzania' has been provisionally accepted for publication in PLOS Global Public Health.

Best regards,

Cynthia Chen

Academic Editor
